# *Beauveria bassiana* (Hypocreales: Clavicipitaceae) Volatile Organic Compounds (VOCs) Repel *Rhynchophorus ferrugineus* (Coleoptera: Dryophthoridae)

**DOI:** 10.3390/jof8080843

**Published:** 2022-08-11

**Authors:** Johari Jalinas, Federico Lopez-Moya, Frutos C. Marhuenda-Egea, Luis Vicente Lopez-Llorca

**Affiliations:** 1Laboratory of Plant Pathology, Department of Marine Sciences and Applied Biology, University of Alicante, 03080 Alicante, Spain; 2Multidisciplinary Institute for Environmental Studies (MIES), University of Alicante, 03080 Alicante, Spain; 3Department of Biological Sciences and Biotechnology, Faculty of Science and Technology, Universiti Kebangsaan Malaysia (UKM), Selangor 43600, Malaysia; 4Department of Agrochemistry and Biochemistry, University of Alicante, 03080 Alicante, Spain

**Keywords:** volatile organic compounds (VOCs), *Rhynchophorus ferrugineus*, *Beauveria bassiana*, insect repellent

## Abstract

The entomopathogenic fungus *Beauveria bassiana* (Bb) is used to control the red palm weevil (RPW) *Rhyncophorus ferrugineus* (Oliver). *Beuveria bassiana* can infect and kill all developmental stages of RPW. We found that a solid formulate of *B. bassiana* isolate 203 (Bb203; CBS 121097), obtained from naturally infected RPW adults, repels RPW females. Fungi, and entomopathogens in particular, can produce volatile organic compounds (VOCs). VOCs from Bb203 were analyzed using gas chromatography-mass spectrometry (GC-MS). GC-MS identified more than 15 VOCs in *B. bassiana* not present in uninoculated (control) formulate. Both ethenyl benzene and benzothiazole *B. bassiana* VOCs can repel RPW females. Our findings suggest that *B. bassiana* and its VOCs can be used for sustainable management of RPW. They could act complementarily to avoid RPW infestation in palms.

## 1. Introduction

The red palm weevil (RPW), *Rhynchophorus ferrugineus* (Coleoptera: Dryophthoridae) is the most serious insect pest for date palms [1,2]. Date (*Phoenix datylifera* L.) and canary palms (*Phoenix canariensis* Chabaud) are economic palms and important in Spain. These species of palms are most abundantly planted in the main tourist resorts in the Mediterranean coastal region from North to South of Spain (Cataluña-Valencia-Murcia-Andalucia).

RPW infestations in Alicante Province (SE, Spain) now affect the UNESCO heritage palms in the cities of Elche and Alicante, Southeastern Spain. In Spain, the economically feasible method to control RPW is by using pesticides and pheromone traps. Excessive use of insecticides causes development of resistance in insect targets and environmental pollution [3]. The mass use of pheromone traps could attract more insects and induce more palm infestations [4]. To minimize the problems caused by current treatments, research interest has been focused on the search for new and environmentally friendly strategies that can provide effective pest control. Thus, understanding the response of RPW behaviour for a specific VOC is essential to find safe solutions to control and prevent RPW spread to palms.

RPW invasion and spread is due to its searching behaviour for food [5,6], mates [7], oviposition and breeding sites [8]. Furthermore, RPW are strong fliers [9] and can climb very tall palms to find resources. These efficient searching mechanisms of RPW are based on antennae that function as chemo- and mechanoreceptors [8]. Volatiles sensed by antennae may alert the insect to the presence of prospective mates, food, suitable places to lay eggs or avoiding chemical dangers. Thus, any chemicals that could interrupt and modify RPW searching abilities may provide a new strategy for RPW control.

Volatile organic compounds (VOCs) are carbon-based solids and liquids that enter the gas phase by vaporizing at 20 °C and 0.01 kPa [10]. VOCs appear as intermediate and end products of various metabolic pathways and belong to numerous structure classes such as mono- and sesquiterpenes, alcohols, ketones, lactones, esters or C8 compounds [10]. Fungi produce various mixtures of gas-phase molecules [11,12,13] which have been shown to be involved in different biological processes such as biocontrol or communication between microorganisms and their living environment. Insect–fungal interactions can be mediated by volatile organic compounds (VOCs) [14].

Fungal VOCs protect stored grains from damage by insects and prevent development of toxicogenic fungi [15]. Insects also perceive and respond to fungal VOCs. Termites with antennae eliminate conidia of entomopathogenic fungi more effectively than insects of the same species without antennae [14]. Fungal VOCs are insecticidal and insect repellents. For example, *Muscodor vitigenus* produces naphthalene, also known as “mothballs,” which are used as insect repellents [16]. Besides, *Muscodor albus* VOCs inhibit *Spodoptera exigua* egg-hatching and are insecticidal [17]. Wheat stem sawfly *Cephus cinctus* is a major pest of wheat that displays avoidance towards *M. vitigenus* [16]. The ability of insects to detect and respond to entomopathogenic fungi within the order Hypocreales has been widely assessed, with reports of avoidance of fungi by species within the Coleoptera [18], Isoptera [14], Hemiptera [19] and Orthoptera [20]. VOCs from fungi showed neurotoxicity effects in *Drosophila melanogaster* [21].

Different species of entomopathogenic fungi such as *Metarhizium anisopliae*, *Metarhizium flavoviride*, *Pandora* sp., *Isaria fumosorosea*, *Hirsutella danubiensis*, *Batkoa* sp. and *Beauveria bassiana* produced a wide profile of secondary metabolites (including VOCs), categorized into groups of aldehydes, ketones, alcohols, esters, acids, terpenes and others [22]. Benzene derivates such as benzaldehyde have also been reported by four species of entomopathogenic fungi (*Batkoa* sp., *Isaria fumosorosea*, *Metarhizium*
*anisopliae* and *Hirsutella*
*danubiensis*) [23]. Recent studies also demonstrate that *B. bassiana* is able to produce VOCs (3-cyclohepten-1-one and 1,3-dimetoxybenze) with repellency activity against banana weevil (*Cosmopolites sordidus*) [24]. The metabolites of entomopathogenic fungi can cause disruption in the normal functioning of insects [25] and some show insect repellent activities [24]. Although some VOCs of entomopathogenic fungi have been explored [12], not all VOCs from entomopathogenic fungi have been described growing on different growth substrates of fungi. A wide and diverse variety of VOCs with different properties have been induced [26,27].

In Spain, the use of a solid formulation of *Beauveria bassiana* from our research group has been successful in reducing the RPW infestation in the field [28]. Pathogenicity of local entomopathogenic fungi isolates against different life stages of RPW have been explored and used in the areas affected by RPW infestation [29,30]. Integration of entomopathogenic fungi and eco-friendly insecticides for management of RPW can reduce the dependent of chemical insecticides [31], avoid insecticide resistance [32] and reduce the economic impact of future management of this invasive pest [33].

Thus, this work investigates the potential of specific VOCs [26,27] from this entomopathogenic fungus as RPW repellent. The hypothesis is that avoidance of the entomopathogen by the insect may be mediated by VOCs. These compounds could be developed as RPW repellents for managing palm infestations in the field, reducing the use of environmentally toxic insecticides. The main objectives in this study were (i) to determine the effect of *B. bassiana* cultures on *RPW* activity, (ii) to identify VOCs produced by solid formulations of *B. bassiana,* and (iii) to investigate the effects of VOCs from *B. bassiana* on *RPW* behavior. This work provides new tools to develop an integrated pest management system to combat RPW palm infestations.

## 2. Materials and Methods

### 2.1. Insects and Palm Petioles Used in the Bioassays

A population of RPW adults were collected in Dolores (Alicante Province, SE Spain) using pheromone traps baited with 4-methy-5-nonanol and 4-methyl-5 nonanone [34]. Insects were maintained in the laboratory in an incubator at 25 ± 0.5 °C in darkness. Plastic boxes (40 × 30 × 21 cm) were set with a folded piece of moistened filter paper containing thin green apple slices that were replaced three times per week [35]. Adults from the stock were sexed by visual inspection of their snouts [36]. Healthy female RPW were randomly selected from the stock population of RPW for Y-tube bioassay experiments, which are explained below.

Palm petioles of *Phoenix dactylifera* were randomly collected at the University of Alicante, (SE Spain). Spines and leaves were removed and the stem was cut into 3 × 2 cm fragments. These cuttings were used for the Y-tube bioassay experiments within 3 h.

### 2.2. Entomopathogenic Fungus, Beauveria bassiana

Entomopathogenic fungus, *B. bassiana* strain used in the experiment, *Bb* 203, was isolated from naturally infected RPW adults in southeast Spain (Daimès, Elche; CBS 121097) [37] and is maintained in the fungal collections of Glen Biotech SL and Department of Plant Pathology, University of Alicante. The fungus is kept in darkness at 4 °C on corn meal agar (CMA; BBL Sparks, MD). The conidia of *B. bassiana* petri plate was prepared and collected according to Assensio et al. (2008) [38]. A solid formulation of *B. bassiana* using rice substrate (*Oryza sativa*) was prepared according to Güerri-Agulló et al. (2010) [28]. 

### 2.3. Description of Y-Tube Olfactometer Bioassay

Y-olfactometers were constructed [39] for laboratory bioassays to test the response of red palm weevil females to volatiles. The olfactometer was constructed entirely from detachable heat-resistant glass tube sections each connected by an airtight cone (5 mm). The glass tubes have an internal diameter of 30 mm, and stem 200 mm and the arms of the Y junction were all 150 mm in length. The angle between each arm and the main body was 75°. The end of each arm was joined to a glass cage (60 mm in length, 25 mm in diameter) in which odor sources were placed. The airspeed inside each arm of the olfactometer was kept constant for all experiments by laboratory air flow. The outlet tubes were covered to prevent the exit of insects (Appendix A). When a tested substance was changed, the jars were washed with sterile distilled water twice [40] and rewashed with n-hexane to remove all remnants of the previous odor [41].

### 2.4. Y-Tube Olfactometer Behavioral Bioassays

Initial experiments for Y-tube bioassays were performed to observe the ability of RPW females to detect and respond to an attractant (palm petiole) or fungus odor in two-choice experiments. Three series of experiments were performed with 20 insects each. These were tested in a Y-tube olfactometer and recorded for 10 min when insects were given a choice between control arm (nothing) and stimulus arm: (i) fresh palm petiole of *P. dactylifera,* (ii) solid formulation of entomopathogenic fungi, *B. bassiana* (iii) the autoclaved rice). Rice (*Oryza sativa*) was used for the two-choice experiments because it was the substrate of the solid formulation of *B. bassiana*. A video camera (Logitech Carl Zeiss Tessar HD 1080P, USA) was used for recording RPW movements in the Y-tube olfactometer. The Y-tube was enclosed within a 40 × 32 × 22 cm plastic box with a window (20 × 10 cm) (Curve plastic Iberia SA) for sample handling.

For each experiment, data obtained on the percentage of responses of adult female RPW to each stimulus versus control arm were subjected to Pearson’s chi-squared (X^2^) analyses to test for significant deviation at *p* < 0.01 from an expected 1:1 (stimulus:control) response.

### 2.5. Effect of the Presence of B. bassiana Solid Formulation on the Behavioural (Y-Tube Olfactometer) Bioassays

*Beauveria bassiana* solid formulation on rice was tested vs. uninoculated rice with RPW females using Y-tube olfactometer as described above. Ten RPW females were recorded individually for 10 min each. The same experiment was repeated using three RPW females but each insect was recorded for 60 min, and mean time spent for RPW in both arms was scored.

### 2.6. Analysis of Volatile Organic Compounds (VOCs) from Entomopathogenic Fungus, B. bassiana 203

One-month-old solid formulations of *B. bassiana* on rice (inoculated with 3 × 10^9^ conidia g^−1^) were used as source for VOCs detection. Tenax in Glass TD Tube 7″ for Gerstel (Supelco, Bellefonte, PA, USA) was used as adsorbent resin for trapping VOCs. Tenax tubes were thermally cleaned with Gerstel thermal desorption system (TDS) (300 °C, 1 h in Nitrogen stream) prior to use. The mass spectra profile (see below) of thermally-desorbed Tenax columns was used as standard blank control.

From ten to twenty grams of solid formulation of the entomopathogenic fungus *B. bassiana* were placed in glass tubes and air flowed for 10 min using an air pump. The end of the glass tube was connected with a thermally cleaned Tenax tube for trapping VOCs from the sample. VOCs from same amounts of uninoculated sterilized rice (control) were collected and trapped as explained. All experiments were repeated three times.

Tenax tubes containing VOCs were subjected to thermal Desorption (TDS) as before and split into the analytic device Gas Chromatography (GC) (Agilent Technologies, 6890 Network GC System) with a HP5-MS 30 m × 0.25 mm × 0.25 μm Column (Agilent, Waldron, Germany). The vaporized VOCs from the sample pass into the ionization chamber of a mass spectrometer (Agilent, 5973 Network Mass Selective Detector). VOCs are identified with Wiley 275 Mass Spectral database by comparison with retention times and spectra with those of known standards.

### 2.7. Behavioural (Y-tube Olfactometer) Bioassays on Candidate Chemicals

In this experiment, ethenyl benzene and benzothiazole identified as main *B. bassiana* VOCs were chosen for Y-tube bioassays to test RPW repulsion activities. The VOCs were tested by two techniques (slow and fast release techniques). Fast chemical release was obtained by injecting 0.5 mL of chemicals at chemical stimuli arm. RPW females (one at time) were released into the opening of the central arm of the Y-tube through and exposed to the two arms: (i) chemical stimuli and (ii) control arms. A total number of 40 healthy RPW females was used for this experiment and 20 RPW were randomly selected for each test with chemical stimuli.

Meanwhile, a modified-method for slow chemical release technique [42] was also applied by using silica gel to trap and slowly release the chemical. This was done by placing 0.5 mL of a given chemical (ethenyl benzene or benzo-tiazole) in 2 g of white silica gel (70-230 mesh, 60 Angstrom, Sigma Aldrich, Saint Louis, MO, USA). Chemical loaded silica-gel was wrapped with cotton wool and placed in the chemical glass cage (60 mm in length, 25 mm in diameter). An individual female RPW was released into the opening of the central arm of the Y-tube with control arm and chemical glass cages when the chemical was inserted into the chemical stimuli arm.

RPW females were tested as for fast release chemical tests (see above). All RPW were given 10 min in the Y-tube to make a choice. A negative response or repulsion behaviour was scored when an insect either moved away from chemical stimuli arm or only stayed in the control arm during the test recording. After each test, RPW was removed and the Y-tube was rinsed with soap, n-hexane (Sigma-Aldrich, Saint Louis, MO, USA), and water and finally dried with tissue and filter paper.

### 2.8. Data Assessment

For each experiment, data obtained on the percentage responses of RPW adult females to each stimulus versus control arm were subjected to chi-square (X^2^) analyses. R studio version 3.1.3 was used to test significant deviation (at *p* < 0.001 from an expected 1:1 (stimulus:control).

## 3. Results

### 3.1. Solid Formulations of the Entomopathogenic Fungus B. bassiana Repel Red Palm Weevil Females

Red palm weevil females (*n* = 20) showed attraction to *P. dactylifera* petioles compared to control arms (X^2^ = 14.2383, df =1, *p* = 0.0001611). Meanwhile the behaviour of RPW females (n = 20) vs. rice and control arms showed no significant differences (X^2^ = 4.0599, df = 1, *p* = 0.04391). Solid formulations of the entomopathogenic fungus *B.*
*bassiana* on rice grains repel RPW females. In a two-choice experiment between the entomopathogenic fungus, *B. bassiana* and control arm, RPW females preferred the control arm instead of that with *B. bassiana* grown in rice (X^2^ = 12.8, df =1, *p* = 0.0003466) (Figure 1).

We also observed the choice of RPW females on uninoculated rice vs. *B. bassiana* formulate. RPW females avoided *B. bassiana* arm and chose the uninoculated rice arm instead (10 min recorded). RPW females recorded for 60 min in a Y-tube avoided *B. bassiana* cultures (0.083 ± 0.06 min) and chose uninoculated rice instead (35.333 ± 13.172 min) (Figure 2).

### 3.2. Identification of B. bassiana VOCs

Volatile organic compounds (VOCs) were likely responsible for RPW repulsion to *B. bassiana* cultures. Therefore, these cultures were subjected to VOCs separation and identification by GC-MS. Analysis with this technique for control-blank samples (Tenax) detected only two main peaks (Figure 3A and Table 1). Seven peaks were found from GC-MS analysis for uninoculated rice (substrate for *B. bassiana* solid formulation) (Figure 3B and Table 2). Eleven peaks were found from GC-MS analysis for 10 g of *B. bassiana* cultures (Figure 3C and Table 3). Finally, 22 main peaks were observed from GC-MS analysis for 20 g samples of *B. bassiana* formulate (Figure 3D and Table 4).

Venn diagram analyses were performed to Tenax only, uninoculated rice and rice fungus formulate samples (Figure 4). These diagrams are very helpful to determine volatiles in common to samples and those unique to each of them. In the Venn diagram we identified four fungal compounds from samples of 10 g of *B. bassiana* formulates (Figure 4A) and 15 compounds from 20 g of this *B. bassiana* formulate (Figure 4B). These compounds were neither present in control (blank) nor in uninoculated rice samples.

VOCs included aromatic compounds, mainly benzene derivates, ketones and hydrocarbons. We identified nine peaks in common from 10 g and 20 g samples of *B. bassiana* formulate (Figure 4C). We chose two candidates from these compounds regarding their potential as repellents. We selected Styrene/Ethenyl-benzene from high volatile and Benzothiazole from low volatile compounds. We used them to run Y-tube bioassay experiments.

### 3.3. Response of Red Palm Weevil Females to Pure VOCs from B. bassiana

Fast release technique*. Rhychophorus ferrugineus* females showed repulsive behaviour during 10 min exposed to either Styrene/Ethenyl-benzene (X^2^ = 14.2383, df = 1, *p* = 0.0001611 or Benzothiazole (X^2^ = 12.9258, df = 1, *p* = 0.0003241) using the fast release technique (Appendix A). RPW females that chose chemical stimuli arms immediately moved away from them, either to control arm or towards the entrance.

Slow-release technique results showed that *R. ferrugineus* females showed repulsive behaviour during 10 min exposure to Styrene/Ethenyl-benzene (X^2^ = 15.0222, df = 1, *p* = 0.0001063) and Benzothiazole (X^2^= 15.5215, df = 1, *p* = 0.00008) in silica gel (Figure 5). It was noticed that RPW females were more comfortable in reaching the middle of Y-tube for making a decision on which y tube arms, compared to the fast release technique.

## 4. Discussion

RPW females avoid the entomopathogenic fungus, *Beauveria bassiana*. Other *Coleoptera* spp. [18,43], *Isoptera* spp. [14], Hemiptera spp. [19] and *Orthoptera* spp. [20] also avoid fungi. The avoidance of female RPW by entomopathogenic fungi (natural enemies) indicate that these weevils could detect danger, especially when finding suitable places for feeding or laying eggs. Entomopathogenic fungi affect oviposition behaviour in the parasitoid wasp *T. rapae* as well, indicating that other insects can also detect danger cues and change their behaviour accordingly [44]. We noticed that *B. bassiana* solid formulate produce an odor different from that of rice (*Oryza sativa*), used as substrate in the formulation [45]. This distinctive odor is due to fungal VOCs which are produced during both primary and secondary metabolism of fungi [12]. Various carbon sources used to grow fungi produce different VOC profiles through metabolic pathways leading to the formation of *B. bassiana* volatiles [11]. For this reason, we analysed both uninoculated rice and *B. bassiana* inoculated rice. In our experiment, we identified VOCs produced by entomopathogenic fungus *B. bassiana* grown on rice. *Beauveriabassiana* produced identified benzene derivatives, benzene acetaldehyde derivatives, straight even-chain saturated hydrocarbons of 10–12 and 16 carbons, *n*-Decane, ketones and alcohol groups. These groups of compounds were also found by Crespo et al., 2008 and other important entomopathogenic fungi such as *Metarhizium anisopliae*, *Metarhizium flavoviride*, *Pandora* sp., *Isaria fumosorosea*, *Hirsutella danubiensis*, *Batkoa* sp. and *Beauveria bassinet* with different number of compositions [22]. Our results showed a positive correlation between amount of fungus formulate and the number of VOCs detected. In this experiment, we found five peaks from both *B. bassiana* samples (10 and 20 g). Styrene/Ethenyl-benzene and Benzothiazole were chosen as low retention time and volatile high retention time volatiles. Both significantly repel RPW females for Y-tube olfactometer assays. Previous studies reveal that styrene reduces attractiveness to pine weevils (both males and females) of pine twig odor [44]. Other studies also confirm that benzene derivatives (benzaldehyde) act as moth repellents [45]. Benzothiazole was reported to have larvicidal and adulticidal activities against mosquitoes [46] and showed a great acute toxic activity against beetle, *T. castaneum* [46]. Recent results also describe that 1,3 dimethoxy-benzene and especially 3-ccycloheptne-1-one synthetized by *B. bassania* can act as important repellents against banana weevil (*C. sordidus*) [24]. Our results show similar repellent activities against *R. ferrugineus*. Repellent compounds such as turmerone, vanillin, nepetalactone and cinnamic acid are repellents of *Tribollium castaneum,* another colleoptera [47]. In our study, RPW females showed repulsion to Styrene and Benzothiazole in Y-tubes using both fast and slow-release techniques. Slow release of chemicals using gel polymers has been used to deliver VOCs such as pheromone [48,49] at a constant rate.

RPW females exposed to *B. bassiana* VOCs had reduced mobility after treatments. This insect behaviour suggests that VOCs could have neurotoxicity. A previous study showed that fungal VOCs had neurotoxicity on *Drosophila melanogaster*, reducing its mobility [21]. In conclusion, our findings suggest that VOCs from *B. bassiana* have potential and can be used to prevent RPW infestation of palms. Future studies should involve further development of gel dispensers for trapping these VOCs, which should allow their use as RPW repellent in the field.

## Figures and Tables

**Figure 1 jof-08-00843-f001:**
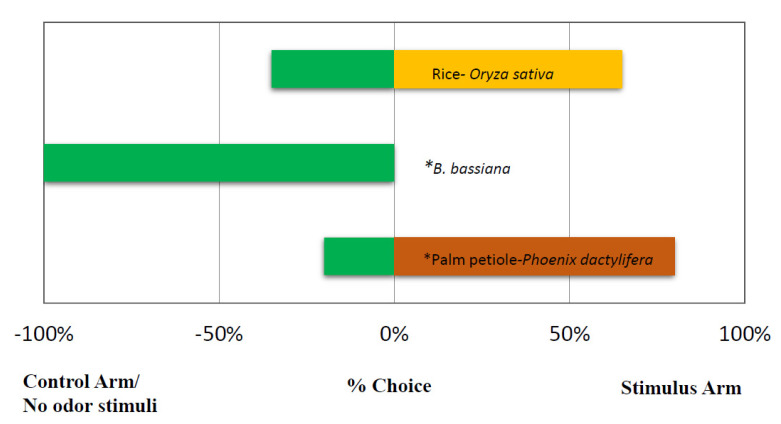
Response of a *Rhynchophorus ferrugineus* female (%) in a Y-tube olfactometer (10 min) when given a choice between environmental air (control) and odor stimulus (host palms and solid formulation of entomopathogenic fungus *B. bassiana*. *N* = 20 individuals per choice test (*p* < 0.01, chi-square). * Indicates significant.

**Figure 2 jof-08-00843-f002:**
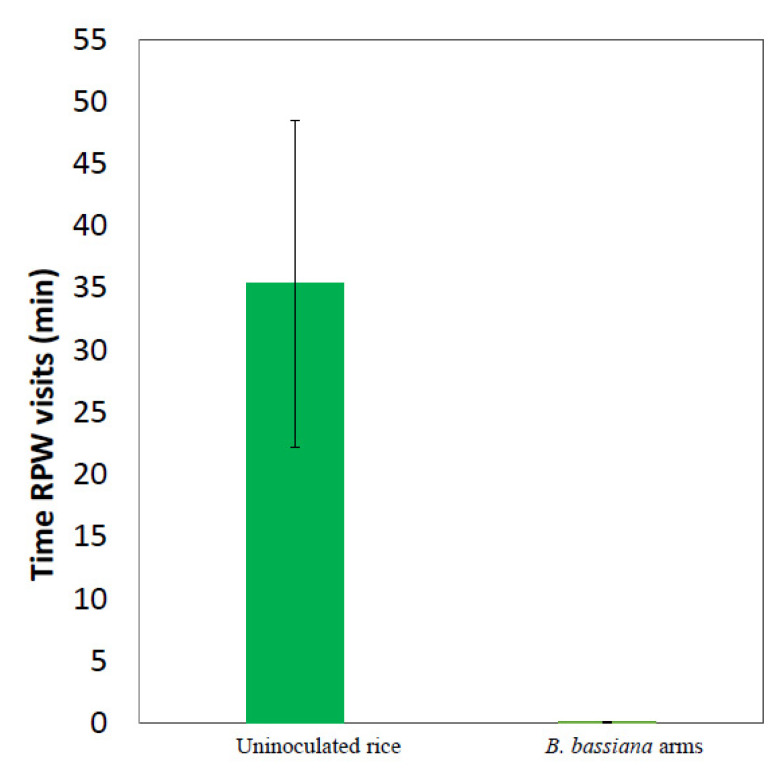
Mean time female RPW (*n* = 3) visits in 60 min recording in Y-tube olfactometer between rice arm and *B. bassiana* arm.

**Figure 3 jof-08-00843-f003:**
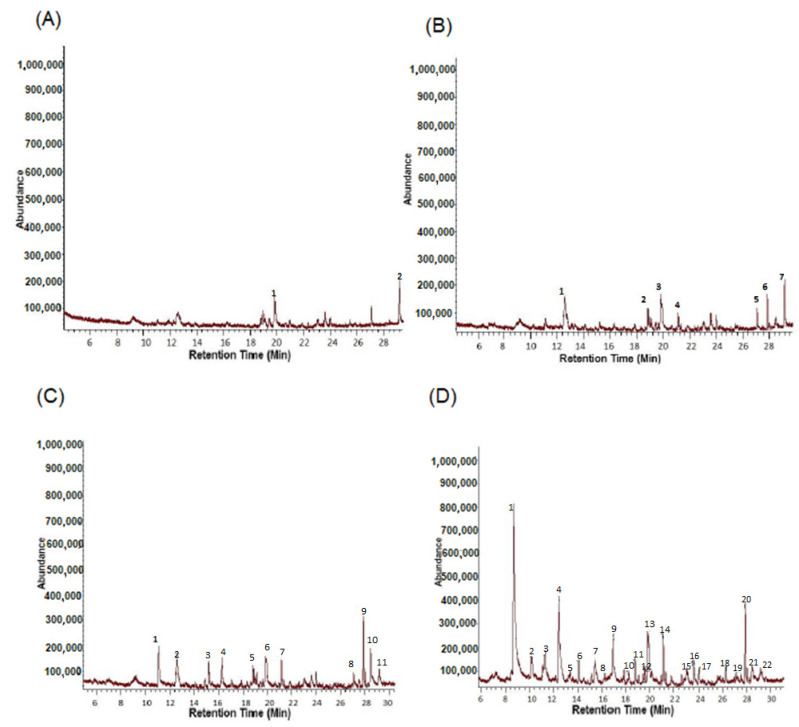
(**A**) GCMS analysis for Tenax (Control). (**B**) GCMS analysis from Rice (*Oryza sativa*) sample. (**C**) GCMS analysis from samples of 10 g of *B. bassiana.* (**D**) GCMS analysis from samples of 20 g of *B. bassiana*.

**Figure 4 jof-08-00843-f004:**
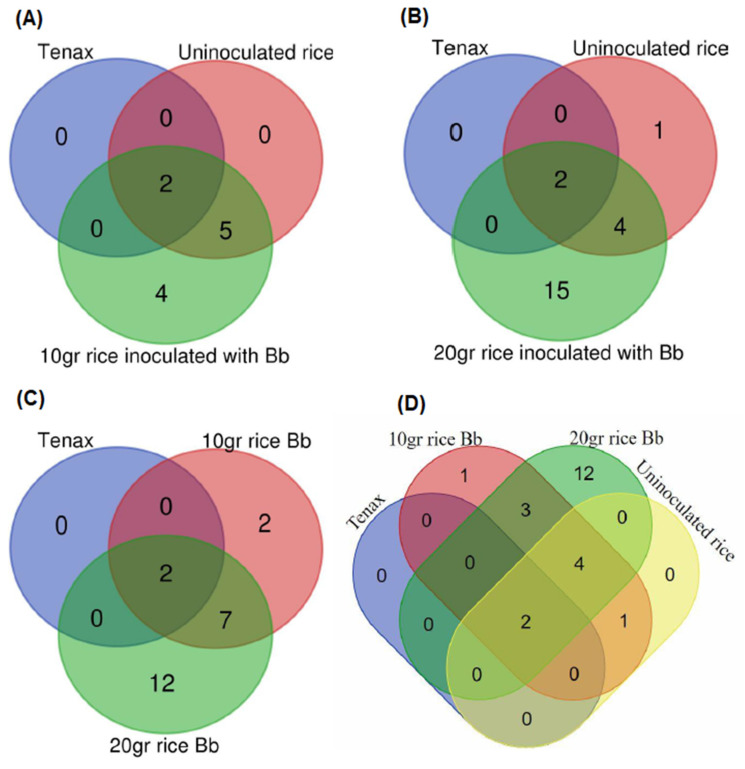
Venn diagram depicting the VOCs detected from the samples. (**A**) blank—green, rice—red, and 10 g of *B.bassiana*—blue. (**B**) blank (green), rice (red), 20 g of *B. bassiana* (blue). (**C**) blank (green), 10 g of *B. bassiana* (red), 20 g of *B. bassiana* (blue). (**D**)**.** Four overlapping samples.

**Figure 5 jof-08-00843-f005:**
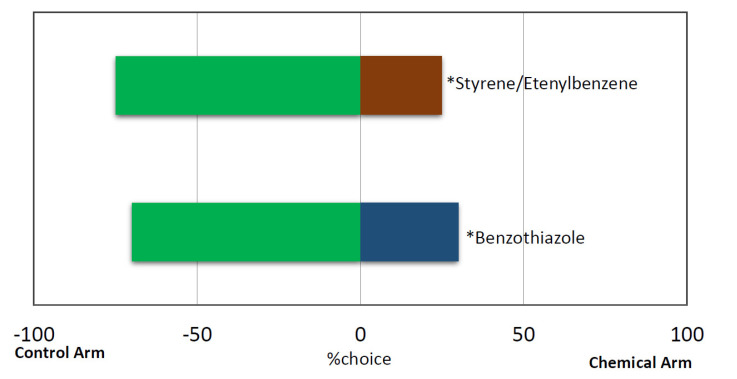
Response of female *Rhynchophorus ferrugineus* in a Y-tube olfactometer (10 min) when given a choice between environmental air (control) and odor stimulus (Chemical stimuli). *N* = 20 individuals per choice test. * Indicates significant difference within a choice test (*p* < 0.01 chi-square).

**Table 1 jof-08-00843-t001:** Volatile organic compounds (VOCs) detected from sample of Tenax (Control).

Peak No.	R.T	Corr. Area	Quality	Compound
1	19.839	6,625,068	93%	Benzaldehyde
2	29.179	6,815,159	91%	1-Decanol

**Table 2 jof-08-00843-t002:** Volatile organic compounds (VOCs) detected from Rice (*Oryza sativa*).

Peak No.	R.T	Corr. Area	Quality	Compound
1	12.564	14,636,750	86%	Cyclotrisiloxane,hexamethyl
2	18.84	3,652,706	86%	Pentasiloxane,dodecamthyl
3	19.843	10,330,336	93%	Benzaldehyde
4	21.137	3,411,857	58%	Undecane, 5-methyl
5	27.099	2,619,884	76%	Capric Aldehyde
6	27.877	5,071,929	87%	Benzene, 1,4-bis(1,1-dimethylethyl)-
7	29.179	8,110,175	91%	1-Decanol

**Table 3 jof-08-00843-t003:** Volatile organic compounds (VOCs) detected from samples of *B. bassiana*.

Peak No.	R.T	Corr. Area	Quality	Compound
1	11.08	9,787,079	93%	Benzene, methyl- (CAS)
2	12.564	12,854,506	86%	Cyclotrisiloxane, hexamethyl
3	15.186	5,433,240	94%	Benzene, 1,4-dimethyl
4	16.27	7,055,705	93%	Styrene/Benzene, Ethenyl
5	18.84	4,447,065	86%	Pentasiloxane,dodecamethyl
6	19.843	10,330,336	93%	Benzaldehyde
7	21.137	5,086,441	58%	Undecane, 5-methyl
8	27.099	2,619,884	76%	Capric Aldehyde
9	27.877	9,834,548	87%	Benzene, 1,4-bis(1,1-dimethylethyl)-
10	28.485	10,858,270	93%	Benzothiazole
11	29.179	8,110,175	91%	1-Decanol

**Table 4 jof-08-00843-t004:** Volatile organic compounds (VOCs) detected from samples of *B. bassiana*.

Peak No.	R.T	Corr. Area	Quality	Compound
1	8.682	76,630,838	86%	Acetic Acid
2	10.202	8,126,923	62%	Hexane, 2,5-dimethyl
3	11.251	8,753,352	50%	1-Butanol, 3-methyl-(impure)
4	12.448	31,223,894	78%	Heptane,2,4-dimethyl
5	14.097	3,768,825	81%	Octane, 4-methyl
6	15.186	5,433,240	94%	Benzene,1,4dimethyl
7	15.463	8,189,342	96%	2-Furancarboxaldehyde
8	16.27	7,055,705	93%	Styrene/Benzene, ethenyl
9	16.942	14,731,012	41%	Disiloxane, pentamethyl
10	18.84	4,447,065	86%	Penatsiloxane, dodecamethyl
11	19.63	2,610,757	46%	Undecane
12	19.843	10,330,336	93%	Benzaldehyde
12a	19.917	9,968,124	86%	Benzaldehyde
13	21.137	8,752,939	58%	Undercane, 5-methyl
14	22.987	2342078	87%	Phenol
15	23.588	5,138,127	95%	Ethanone. 1-phenyl
16	24.01	3,254,487	64%	Cyclopentasiloxane, decamethyl
17	26.250	4,020,849	46%	3-Hexyn-1-ol
18	27.099	2,619,884	76%	Capric Aldhyde
19	27.877	11,251,518	87%	Benzene, 1,4-bis(1,1-dimethylethyl)-
20	28.485	4,952,743	93%	Benzothiazole
21	29.179	8,110,175	91%	1-Decanol

## Data Availability

Not applicable.

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
