# Peer review of "Beauveria bassiana (Hypocreales: Clavicipitaceae) Volatile Organic Compounds (VOCs) Repel Rhynchophorus ferrugineus (Coleoptera: Dryophthoridae)"

_jof, 2022, doi:10.3390/jof8080843_

Round 1
Reviewer 1 Report
The manuscript has been improved and every comment was taken into consideration,
Author Response
Dear Reviewer,
The authors are very much thankful to the reviewers for their meticulous review and valuable comments. Please, you can check all the changes suggested in the new version of the manuscript. We indicate each point raised by the reviewer (located with page and line nos. of our previous manuscript and indicated in bold and italics under response) together with an explanation of our amendments or corrections in the new version of the manuscript. We hope that the revised manuscript is now suitable for publication in Journal of Fungi Journal. We wish to thank the technical criticisms made to our manuscript, as well as the efforts spent on grammar and style of the text corrections. They have helped us to improve this new version of the manuscript. Many thanks in advance.

Reviewer 2 Report
Title: Beauveria bassiana (Hypocreales: Clavicipitaceae) Volatile Organic Compounds (VOCs) Repel Rhynchophorus ferrugineus (Coleoptera: Dryophthoridae)
The current manuscript has a good objective, and as it is known about Beauveria bassiana, it has an effective role in the biological control of many insects. In this research, components produced from this fungus were analyzed as secondary products that have an effect on female insects, which would reduce the number of harmful insects.
Abstract:
-it is good, but it short, the authors should consider the proposed changes for improving the clarity of the content. Such add the background on Beauveria bassiana and its effects and harmful of Rhynchophorus ferrugineus
Keyword: good
-Introduction part is appropriate but a few things are needed for further improvements especially the study aims should be added. Update the references
Add some studies about the study with highlighting research gaps, which necessitated conducting this trial.
Materials and methods:
-this part describes very well by using suitable subheadings. However, it needs few modifications and details of selecting primers and amplification conditions in the revised version to enhance
Need details about A solid formulation of B. bassiana using rice substrate (Oryza sativa)
Results and Discussion
-Both parts need to combine and it needs major revision and it needs some figs of treated insects and show the effects of the fungus on them
Some tables are repeated why???
Figures 3 and 4 need draw with different stile to be more clear for readers
Conclusion:
not included
References:
-Cross-check the references in the text and reference cite. Few references are not as per journal style in the text as well reference section
Author Response
Dear Reviewer,
The authors are very much thankful to the reviewers for their meticulous review and valuable comments. Please, you can check all the changes suggested in the new version of the manuscript. We indicate each point raised by the reviewer (located with page and line nos. of our previous manuscript and indicated in bold and italics under response) together with an explanation of our amendments or corrections in the new version of the manuscript. We hope that the revised manuscript is now suitable for publication in Journal of Fungi Journal. We wish to thank the technical criticisms made to our manuscript, as well as the efforts spent on grammar and style of the text corrections. They have helped us to improve this new version of the manuscript. Many thanks in advance.
Corresponding Author

Reviewer 3 Report
Please add some experimental figures.
Add new studies in the introduction part, likewise the articles published in 2020 -2022.
Author Response
Dear Reviewers,
The authors are very much thankful to the reviewers for their meticulous review and valuable comments. Please, you can check all the changes suggested in the new version of the manuscript. We indicate each point raised by the reviewer (located with page and line nos. of our previous manuscript and indicated in bold and italics under response) together with an explanation of our amendments or corrections in the new version of the manuscript. We hope that the revised manuscript is now suitable for publication in Journal of Fungi Journal. We wish to thank the technical criticisms made to our manuscript, as well as the efforts spent on grammar and style of the text corrections. They have helped us to improve this new version of the manuscript. Many thanks in advance.
Corresponding Author

This manuscript is a resubmission of an earlier submission. The following is a list of the peer review reports and author responses from that submission.
Round 1
Reviewer 1 Report
This paper entitled"Beauveria bassiana (Hypocreales: Clavicipitaceae) Volatile Organic Compounds (VOCs) Repel Rhynchophorus ferrugineus (Coleoptera: Dryophthoridae)" is appeared to be a nice piece of work and will provide more information and reference for future study. However, I think this need major revision to accept. The main problems were as follows:
1. The VOC of Beauveria bassiana and other entomopathogenic fungi have been studied. I strongly recommend the authors to add some related infoumation in the Introduction and Discussion.
2. Crespo et al. (2008) reported the volatile organic compounds released by Beauveria bassiana. Compared with the kind of VOC, whether the substrates could affect the generation of VOC?
3. I would like to recommend the authors to increase the positive control group and compare it with other insect repellents.
Author Response
Dear reviewer,
you can find enclosed our response to your comments and suggestions.
Many thanks

Reviewer 2 Report
This manuscript describes the identification and use of two VOCs produced by Beauveria bassiana able to repel RPW insects. This research is interesting for the possible control of insect pest, although, the work could be substantially improved if some missing details are presented, in addition to comparison with similar results. There are some critical points that need revision to find the novelty of the study as listed below.
Lines 19 and 20: Avoid the use of "low" and "high volatile" this could lead to further confusions
Lines 30-40: What are the advantages relative to volatile hormones?
Line 47: Verify the final hyphen.
Line 62: Check the double "is is" in the phrase
Lines 69-70: The hypothesis as written has already been proven, as mention above. Try to state the novelty of this work. There are not data concerning the concentration of VOC used and if those concentration are similar to those produced bb Beauveria bassiana in the cultures.
Lines 71-72: There are some errors in the typography, including the use of italics fond randomly. Watch the writing of "Bbassiana"
Line 78: Do these pheromones attract randomly female and male insects?
Line 89: How was this strain identified?
Lines 93-94: Briefly give more details on the formulation. Were spores first collected? Was fungal biomass used directly from a solid culture?
Later it is confusing to establish whether solid culture and formulation cultures refer to the same concepts.
It is not possible to know if a conventional solid culture was done neither under what conditions.
Line 130: It seems that one month old cultures (or formulations) are not suitable to VOCs detection. Why did you choose such a time?
Were these time criteria used in the Y-tube olfatometers? Please explain.
Line 131: Use international convention for unit abbreviations (grams=g)
Line 137: "Ten-twenty"?
Lines 150 and others: The use of "high" or "low" volatiles is a rather informal manner to refer to chemical compounds. Did you refer to molecular weight or chain length?
Watch the use of Bold fond in next lines.
Line 153: Concentration of chemicals? Equivalent to that found in the formulations?
Tables: Insert the table legends in those places close to the corresponding tabla, this would help the reader to analyse the results.
Table 4: The compounds were not novel molecules for entomopathogenic fungi. Why not used them in combinations?
Lines 229-233: The compounds occur simultaneously in fungal cultures. Why were they tested only separately and not tested jointly? This would be a stronger novelty of the work.
Line 264: Avoid abbreviations after a period or at the beginning of a paragraph (".B. bassiana”)
Line 277: How was mobility assessed?
Author Response

(The authors gave the same response as above.)

Round 2
Reviewer 1 Report
The revised paper has been improved. However, the problems mentioned last time are still unsolved, such as recommend the authors to add some related infoumation in the Introduction and Discussion and increase the positive control group and compare it with other insect repellents. Besides, the VOCs were commonly present in entomopathogenetic fungi. Unfortunately, the introduction and disscussion were Less involved. Thus, I strongly recommend the authors to revise the paper carefully and compared with other entomopathogenetic fungi. Thus, I would like suggest the Editor to reject the paper in the present form and encouraged resubmission after revision manuscript.
Reviewer 2 Report
The manuscript has been sufficiently improved to recommen publication in JoF.